# Interleukin-15 in Outcomes of Pregnancy

**DOI:** 10.3390/ijms222011094

**Published:** 2021-10-14

**Authors:** Scott M. Gordon

**Affiliations:** 1Division of Neonatology, Children’s Hospital of Philadelphia, Philadelphia, PA 19104, USA; gordons1@chop.edu; 2Department of Pediatrics, Perelman School of Medicine, University of Pennsylvania, Philadelphia, PA 19104, USA

**Keywords:** Interleukin-15, CD122, pregnancy, inflammation, placenta, natural killer cells, macrophages, trophoblast

## Abstract

Interleukin-15 (IL-15) is a pleiotropic cytokine that classically acts to support the development, maintenance, and function of killer lymphocytes. IL-15 is abundant in the uterus prior to and during pregnancy, but it is subject to tight spatial and temporal regulation. Both mouse models and human studies suggest that homeostasis of IL-15 is essential for healthy pregnancy. Dysregulation of IL-15 is associated with adverse outcomes of pregnancy. Herein, we review producers of IL-15 and responders to IL-15, including non-traditional responders in the maternal uterus and fetal placenta. We also review regulation of IL-15 at the maternal–fetal interface and propose mechanisms of action of IL-15 to facilitate additional study of this critical cytokine in the context of pregnancy.

## 1. Introduction

The regulation of pro- and anti-inflammatory signals is a critically important aspect of pregnancy, from implantation of the embryo to delivery of a newborn. As in countless other contexts, immune homeostasis must be maintained at the maternal–fetal interface to ensure the health of the mother and fetus. Both inappropriate immune activation and inappropriate immune quiescence are associated with adverse outcomes of pregnancy.

Multiple cytokines and chemokines regulate the unique composition and function of immune cells at the maternal–fetal interface. Interleukin-15 (IL-15) is a pleotropic cytokine that classically supports development, maintenance, and activity of killer lymphocytes [1,2,3,4,5]. Populations of natural killer (NK) cells, NKT cells, and memory CD8+ T cells are severely impaired in the absence of IL-15 in mice. Prior to pregnancy and early in gestation, NK cells are evident in the uterus in staggering concentrations relative to other lymphoid and non-lymphoid organs. While not all functions for NK cells in pregnancy are known, it has become clear that NK cells play key roles in healthy and complicated gestations [6]. Given the abundance and importance of NK cells in pregnancy, detailed study of IL-15 is essential.

The signaling components of the IL-15 receptor complex have been well described [7,8]. In brief, the IL-15 receptor complex is heterotrimeric, consisting of an α chain (CD215), a β chain (CD122), and the common γ chain (γc/CD132) (Figure 1). IL-15Rα binds IL-15 with high affinity and presents it to cells expressing high levels of CD122 and γc. Presentation of IL-15 may occur in cis, with the IL-15/IL-15Rα complex on the same cell as CD122/γc, or presentation of IL-15 may occur in trans, with IL-15/IL-15Rα on one cell and CD122/γc on another [5,9,10,11].

Of note, CD122/γc also represents the intermediate-affinity IL-2 receptor complex. Despite the fact that IL-2 and IL-15 share key components of their respective receptor complexes, we focus here on IL-15. IL-2 is completely absent from the maternal–fetal interface during steady-state pregnancy in mice. In a study of cytokine dynamics during mouse pregnancy, whole uterine tissue, plus placenta later in gestation, was subjected to qRT-PCR at nearly every day of gestation [12]. No *Il2* transcript was detected. In a large, publicly available dataset of single-cell RNA sequencing (scRNAseq) of hematopoietic and non-hematopoietic cells during the first trimester of human pregnancy, *IL2* transcript is also absent from the decidua and placenta [13].

## 2. Spatial and Temporal Regulation of IL-15 in the Uterus

Like many other inflammatory cytokines, IL-15 is regulated at multiple levels to maintain homeostasis. Human *IL15* transcript exists as two isoforms, one with a long signal peptide sequence (LSP) and one with a short signal peptide sequence (SSP) [14,15]. When translated, the resultant IL-15 proteins are differentially localized in the cell and differentially secreted [15]. Soluble, free IL-15 is evident in the serum of mice in the steady state, as is small but nonzero amounts of IL-15 complexed with IL-15Rα [16]. Both isoforms of IL-15 are stabilized by IL-15Rα and exhibit greatly enhanced bioactivity on target cells when bound to IL-15Rα in vitro and in vivo [15,17,18,19]. The IL-15/IL-15Rα complex may remain bound to the cell membrane to spatially restrict the activity of IL-15 only to target cells directly contacting the IL-15-presenting cell. Instead, IL-15/IL-15Rα may be cleaved by a variety of mechanisms from the cell surface to act on more distant target cells [20].

Both *Il15* and *Il15ra* transcripts in mice have been detected by qRT-PCR in bulk uterus alone in early gestation or in bulk uterus and placenta later in gestation [12]. *Il15* is induced during the preimplantation period, by embryonic day 3 (E3), and peaks during mid-gestation, from E8 to E10 (Figure 2). It declines thereafter to a lower but constant level through E18.5. With *Il15*, *Il15ra* is also strongly induced from E8 to E10, but it is nearly undetectable in bulk tissue at every other gestational day. These were semi-quantitative and neither completely rule out low expression of *Il15ra* nor rule out the persistence of IL-15Rα protein even after production of new transcripts ceases. However, uterine NK (uNK) cells in mice, identified by the binding of *Dolichus biflores* agglutinin (DBA), expand and contract with the same kinetics as the *Il15ra* transcript. In a kinetic, flow cytometric study of conventional NK1.1+DX5+NK cells and DBA+ uNK cells, uNK cells are rare prior to pregnancy and at E6.5, expand at least 10-fold and peak at E9.5, and return nearly to pre-pregnancy levels by E13.5 [21]. These data support a model in which IL-15 is induced just prior to implantation of the embryo in mice, peaks at mid-gestation around the time of spiral artery remodeling, and declines but persists to term. Uterine NK cells may then differentiate and expand in response to IL-15/IL-15Rα, but not to IL-15 alone (Figure 1).

In the human uterus, the first studies to characterize IL-15 at the maternal–fetal interface did so with semi-quantitative RT-PCR and immunohistochemistry (IHC) [22,23]. With endometrial tissues collected after hysterectomy for leiomyomas or carcinoma in situ of the cervix, work by Kitaya found both the SSP and LSP isoforms of *IL15* transcript evident at low levels during the proliferative phase [22]. Both SSP and LSP *IL15* were induced during the secretory phase, and in first trimester elective terminations of 7–11 weeks gestation, *IL15* transcripts remained elevated (Figure 2). Data by Verma agree with low levels of *IL15* in proliferative phase endometrium [23]. While Verma found minimal to no induction of *IL15* transcript in secretory phase endometrium, at odds with the data by Kitaya, the highest levels of *IL15* transcript were found in pregnant decidua. Different findings between these two studies may reflect differences in when, exactly, tissues were obtained during the secretory phase. In recent studies of human endometrial tissue, exact dating was possible during proliferative and secretory phases [24,25,26]. With such granular data, it was shown that the *IL15* transcript is induced only during the second half of the secretory phase, which correlates to the window of implantation, or the point in the menstrual cycle, during which the uterus is most ready to receive an embryo (Figure 1) [24,25,26,27].

In the study by Kitaya, IHC for IL-15 protein in the proliferative phase endometrium showed staining in endometrial glandular epithelial cells, with very weak stromal cell staining [22]. IL-15 staining increased throughout the stroma and was most prominent in perivascular cells around spiral arteries in the secretory phase endometrium (Figure 2). In first-trimester decidua, IL-15 was found throughout the stroma, as well as in vascular endothelial cells. Newer data confirm the induction of *IL15* in stromal cells of the secretory phase endometrium by RNA in situ hybridization, as well as IL-15 staining by IHC in pregnant decidua [24,25]. The data by Verma add that the *IL15* transcript was abundant in decidual macrophages enriched by adherence to tissue culture plastic [23]. While these data agree with scRNAseq data showing decidual macrophages at the first-trimester maternal–fetal interface express *IL15* [13], decidual stromal cells are also adherent and strongly express *IL15* (as discussed below), perhaps biasing detection of IL-15 transcript in this assay.

By flow cytometry, surface IL-15, presumably bound by IL-15Rα, was detected on CD14+ and CD14− cells (Figure 2) [22]. CD14+ cells in the uterus most likely represent monocytes, macrophages, and/or monocyte-derived dendritic cells. The identity of the CD14− cells as hematopoietic or non-hematopoietic was not revealed, as flow data were not gated on CD45+ cells. As most decidual stromal cells express a large amount of *IL15* and *IL15RA* [13], we presume that these CD14− cells presenting IL-15 are decidual stromal cells. While Kitaya showed that IL-15 was not detected on the surface of CD56^bright^ NK cells in first-trimester decidua [22], Verma reported the *IL15RA* transcript in CD56^bright^ cells purified by magnetic cell separation [23]. While relatively few NK cells in first-trimester human decidua express *IL15* and/or *IL15RA* by scRNAseq [13], NK cells can express IL-15Rα and signal in cis in other contexts [11].

## 3. Hormonal and Transcriptional Regulation of IL-15 in the Uterus

Taken together, the data discussed above support that the uterus expresses a modest amount of IL-15 until the late secretory phase, after the endometrial lining has remodeled, or decidualized [27,28,29], in preparation for an embryo (Figure 2). The uterus remains rich in IL-15 through the first trimester of pregnancy. IL-15 levels then wane and are accompanied by a withdrawal of IL-15Rα, which would limit trans-presentation of IL-15 and could lead to contraction of the killer lymphocyte compartment. Contributions by reproductive hormones may shed additional light on the regulation of IL-15 at the maternal–fetal interface.

Estrogen is far in excess of progesterone until the beginning of the secretory phase, when progesterone levels begin to rise and the process of decidualization begins [27,28,29]. The timeframe during which levels of progesterone become dominant represents the window of implantation. One interpretation of the data presented thus far is that an estrogen-dominant hormonal environment restricts expression of IL-15. Another interpretation is that progesterone needs to reach a critical level before it can drive the expression of IL-15. Alternatively, progesterone may initiate a cascade of events that culminates indirectly, over days, in the induction of IL-15. In other words, expression of IL-15 may be a distal feature of the decidualization program (Figure 2).

Kitaya cultured matched samples of bulk decidual cells for 5 days in medium alone or in medium plus progesterone [22]. Supernatants contained inconsistently higher concentrations of IL-15 after 5 days of exposure to progesterone. Effects of estrogen were not tested. Verma found that cultures enriched for decidual macrophages produced over twofold less IL-15 by ELISA after exposure to progesterone and prostaglandin E2 (PGE2) [23]. This same medium containing progesterone and PGE2 stimulated decidual stromal cells to produce about 50% more IL-15 by ELISA. These data suggest that hormonal control of IL-15 production is complex and may be cell type-specific.

More recently, two groups have examined the role of the transcription factor Heart and neural crest derivatives-expressed transcript 2 (HAND2) in regulating IL-15. Shindoh studied the effects of HAND2 on expression of IL15 in Vimentin+ primary endometrial stromal cells (ESCs) cultured for 12 days in estrogen and progesterone to decidualize them [30]. Knockdown of HAND2 with siRNA resulted in loss of multiple decidualization-associated genes, including IL-15 (Figure 2). More recently, the same group showed coordinate induction of HAND2 transcript and protein, as well as IL15 transcript and protein, in secretory phase endometrium [24]. Of note, not every stromal cell expressed IL-15 protein. IL-15 expressers appeared to be evenly distributed throughout the stroma. These data suggest that certain stromal cells may be specialized in their ability to express IL-15. Instead, IL-15 expression may be stochastic.

To answer whether HAND2 had a direct effect on IL-15 or a secondary effect by virtue of controlling decidualization, Murata found a HAND2 binding motif conserved between mice, macaques and humans at position -1628 to -1622 relative to the IL15 transcriptional start site (TSS) [24]. By CHIP-qPCR, HAND2 was found to bind this site of the IL15 promoter. Next, ESCs were decidualized in culture and transfected with luciferase constructs containing the *IL15* promoter region with the HAND2 motif intact or mutant versions of the *IL15* promoter region lacking the intact HAND2 binding motif. Co-transfection of these constructs with a HAND2 expression vector drove expression of luciferase only in the presence of the intact promoter region.

Similar to findings by Murata, Marinić recently found that levels of *HAND2* and *IL15* transcript were directly correlated in preparation for and during pregnancy [25]. They added a complete kinetic of levels of both *HAND2* and *IL15* over the course of human gestation, showing induction during the mid-to-late secretory phase into the first trimester of pregnancy and showing a relative decrease in levels of *HAND2* and *IL15* transcripts in the basal plate, or the maternal side, of the placenta as gestation approached term (Figure 2). Broadly speaking, these human data agree with the kinetics of murine *Il15* during gestation, but expression of IL-15Rα was not examined. In contrast to prior findings [24,30], however, Marinić found that HAND2 negatively regulated *IL15* using siRNA knockdown in immortalized endometrial stromal fibroblasts (ESFs) [25]. Marinić used the hTERT-immortalized cell line CRL-4003, derived from mid-secretory ESFs, and performed siRNA knockdown of HAND2 for 48hrs [25,31]. Murata and Shindoh used bulk Vimentin+ primary ESCs, which likely contain ESFs and more differentiated DSCs [32], and performed siRNA knockdown in the presence of decidualization hormones for 12 days. So, the discrepant results are likely due to differences in cell type, timing, and differences in culture conditions.

Altogether, these data suggest that HAND2 directly binds the *IL15* promoter and regulates expression of IL15 (Figure 2). HAND2 clearly can bind a motif proximal to the *IL15* TSS [24]. Whether HAND2 promotes or represses transcription may depend on cell type/state, with activation seen in decidualized primary ESCs [24] and repression seen in undifferentiated, immortalized ESFs [25]. It is possible, though it remains to be formally shown, that HAND2 also directs expression of *IL15* by binding to more distant enhancers. In silico analyses show that HAND2 binding motifs exist beyond the *IL15* promoter, within distant enhancers that contact the *IL15* promoter [25].

Additional factors that may drive expression of *IL15* were identified by analysis of publicly available siRNA knockdown data in primary ESFs differentiated to DSCs with cAMP and progesterone. Consistent with the notion that IL-15 is part of a decidualization program (Figure 2), *IL15* was induced by differentiation of ESFs into DSCs with cAMP and progesterone [25]. Knockdown of *PGR*, encoding the progesterone receptor, diminished the *IL15* transcript approximately twofold, consistent with data that the progesterone receptor antagonist asoprisnil suppresses *IL15* [26]. Knockdown of GATA Binding Protein 2 (GATA2) diminished the *IL15* transcript over fourfold [25]. Of potential interest is that GATA2 has been implicated in differentiation and maintenance of CD56^bright^ NK cells in humans, the precursors of more mature CD56^dim^ NK cells and the predominant phenotype of human uNK cells [33]. While GATA2 mutations appear to cause NK cell-intrinsic defects, it was not formally shown whether these inborn errors of immunity also impact the IL-15 axis, with a secondary effect on NK cell differentiation and/or survival. In the context of myelopoiesis and lymphopoiesis in the steady state, expression of IL-15 using a fluorescent reporter was demonstrated in hematopoietic progenitors and myeloid cells, the transcriptional regulators of which remain unknown [34]. The same group showed that upregulation of IL-15 occurred after viral infection and that induction of IL-15 was dependent on the type I interferon (IFN) receptor IFNAR [35]. Type I IFNs are present in the uterus at low levels in the steady state and contribute to host defense [36], but whether IFNs play a role in decidualization and/or induction of IL-15 in the uterus has not been examined.

## 4. Spatial and Temporal Regulation of IL-15 in the Placenta

While IL-15 is rich in the maternal uterus during early gestation, the developing fetal placenta is relatively poor in IL-15 early in gestation. Minimal *IL15* transcript was found in flow-sorted, first-trimester epidermal growth factor receptor-positive (EGFR+) villous trophoblasts (VTs) and erbB2+ extravillous trophoblasts (EVTs) [23], consistent with scRNAseq data mentioned previously [13]. Toth shows minimal expression of IL-15 in multiple cells of the normal first-trimester placenta by IHC, including EVT and syncytiotrophoblast [37]. Agarwal measured IL-15 by ELISA and *IL15* by semi-quantitative RT-PCR in cultured human placental explants [38]. While these were bulk explants containing a heterogenous population of cells, production of IL-15 increased twofold from first-trimester explants to full-term explants from spontaneous vaginal deliveries. Full-term explants from elective cesarean sections in the absence of labor produced an intermediate amount of IL-15. Placental macrophages isolated at term and cultured for 24 h produce an undetectable amount of IL-15 by ELISA [39]. These limited data suggest that the human fetal placenta is capable of making progressively more IL-15 over the course of gestation, with a peak during labor (Figure 2). Trophoblasts, not macrophages, appear to be responsible for the production of IL-15 in the placenta.

## 5. Dysregulation of IL-15 in Adverse Outcomes of Pregnancy

Multiple lines of evidence in mice and humans suggest that either excess or loss of IL-15 leads to adverse outcomes of pregnancy. First, a genetic deficiency of IL-15 in mice leads to fetal growth restriction, morphologically abnormal deciduae, and impaired remodeling of spiral arteries [40,41], a pathologic hallmark of preeclampsia. IL-15 knockout dams were also shown to have a modestly but consistently elevated rate of fetal resorption during pregnancy, compared to IL-15-sufficient dams [42]. However, IL-15-deficient dams mated to IL-15-deficient sires still have a normal length of gestation and deliver a normal number of viable pups [40,41]. Both groups to report this phenotype demonstrate an absence of DBA+ uNK cells, consistent with the notion that IL-15/IL-15Rα drives differentiation of uterine-phenotype NK cells, as discussed above (Figure 2). Of note, DBA-negative NK cells in the uterus were not examined and may support gestation in important ways. More broadly, IL-15-independent NK cells in the uterus have not been examined. It has been shown by two different groups that interleukin-12 can support the development and expansion of NK cells independently of γc cytokines [43,44]. It is also important to appreciate that relatively minor adverse outcomes of pregnancy in IL-15-deficient mice and in other NK cell-deficient mice were observed in the setting of syngeneic matings [40,41,45,46]. Despite the hardiness of syngeneic mouse pregnancy, fetal growth restriction and preeclampsia are of major clinical significance in humans. In allogeneic matings, *Nfil3*/E4BP4-deficient mice that have a severe reduction in NK cells also exhibit substantial fetal loss [47].

Overabundance of IL-15 in mouse models is also associated with poor outcomes of pregnancy. IL-15 permits fetal loss in a model of lipopolysaccharide (LPS)-mediated abortion, in which low-dose LPS is given to pregnant dams post-implantation at E7.5 [42]. In other words, IL-15-deficient dams are completely resistant to fetal loss in this system. Whether IL-15 permits inflammation-mediated fetal loss by activating NK cells was not formally shown. With an anti-NK1.1 antibody known to deplete NK1.1+ NK cells in vivo, though, the authors demonstrated partial rescue of fetal loss in response to LPS. The mechanism of this remains unclear, as successful depletion of NK cells in the uterus was not shown, and NK1.1-DBA+ uNK cells should not be depleted using this method [21].

Another example of unrestrained IL-15 is in the BPH/5 mouse, a model of spontaneous hypertension and preeclampsia [48,49,50]. Compared to control animals, pregnancies in BPH/5 dams are characterized by implantation defects, placental abnormalities, fetal growth restriction, and fetal loss. Despite having abnormal, delayed decidualization, BPH/5 implantation sites exhibit inappropriately elevated IL-15 transcript and protein early in gestation, during the peri-implantation period and through at least E7.5 [48]. Thus, decidualization and IL-15 are uncoupled in this model. Levels of IL-15 normalize in BPH/5 implantation sites by E10.5, when IL-15 peaks in normal mouse pregnancy. Paradoxically, however, DBA+ uNK cells are decreased in BPH/5 mice by flow cytometry, NK cell-related transcripts, including *Ncr1*, and by immunofluorescence of E7.5 implantation sites. It is worth noting that uNK cells are rare early in gestation [21], and the authors’ raw data confirm this [48]. At the same time, the number of CD122+TCRβ- lymphocytes, presumably conventional DBA- NK cells, appears to be severely reduced by flow cytometry in the BPH/5 mouse. Treatment of pregnant WT mice with recombinant IL-15 precomplexed with IL-15Rα also reduced DBA+ NK cells- and *Ncr1* transcript in early implantation sites. Administration of the cyclooxygenase 2 (COX2) inhibitor celecoxib at E6.5 reduced resorption of fetuses, improved fetal-placental weights, and quickly (by E7.5) normalized IL-15 and NK cells back to WT levels. These data support that IL-15 must be tightly regulated in a spatial and temporal manner during pregnancy and that the relationship between uNK cells and IL-15 is complex and not fully understood.

One caveat to applying studies of IL-15 in mice is that mouse and human cells differentially depend on, and respond to, IL-15. A single 10 μg intraperitoneal dose of an antibody raised against murine IL-15 results in near-total elimination of classical splenic DX5+NK1.1+ NK cells in less than 5 days [51]. Blood NK cells also sharply declined in cynomolgus macaques administered 5 weekly doses of intravenous or subcutaneous anti-IL-15 antibody by 2 weeks post-treatment through 20 weeks post-treatment. Despite high levels of circulating IL-15-blocking antibody in healthy volunteers, neither CD56^bright^ nor CD56^dim^ NK cells in the blood declined in number during the study. These data suggest that, in contrast to murine and non-human primate NK cells, human NK cells do not depend on IL-15. While they may not critically depend on IL-15, human NK cells are at least partially dependent on Janus kinase 3 (JAK3), which directly associates with γc. A safety and efficacy study in kidney transplant recipients showed that a JAK3 inhibitor (that also modestly inhibits JAK2) substantially decreases the absolute number of circulating NK cells [52]. Investigations of IL-15-dependent and -independent mechanisms of uNK cell development and function may help to reconcile these differences between mouse and human studies and refine our understanding of roles for IL-15 in human pregnancy.

Further study of IL-15 is essential, as IL-15 is consistently dysregulated in the setting of adverse outcomes of pregnancy in humans. Placental explants from term pregnancies affected by severe preeclampsia produce fourfold less IL-15 in culture than explants prepared from healthy term pregnancies [38]. In the uterus of women with preeclampsia, *IL15* transcript was upregulated tenfold in DSCs but only twofold in ESFs [25]. IL-15 was modestly elevated in the serum of women with preeclampsia at term [53]. In the uterus of women with implantation failure, *IL15* transcript was downregulated over twofold and trended toward marginal downregulation in women with recurrent spontaneous abortion. IL-15 protein by IHC in the placenta is elevated in the setting of miscarriage and further elevated in the setting of recurrent miscarriage [37]. These data underscore that adverse outcomes of pregnancy are associated with disturbed homeostasis of IL-15 and that IL-15 is regulated (and dysregulated) in a cell type-specific manner. Whether abnormal homeostasis of IL-15 is a cause or effect of adverse outcomes of pregnancy, and whether targeting IL-15 might reverse such conditions and restore maternal–fetal health, remain to be investigated.

## 6. Traditional and Non-Traditional Responders to IL-15

Effects of IL-15 on NK cells, as well as signaling and metabolic cascades activated in response to IL-15, have been extensively studied and are the subject of several comprehensive reviews [54,55,56]. In brief, IL-15 supports NK cells in many ways, including the development, maintenance, cytolysis, antibody-dependent cell-mediated cytotoxicity, and production of cytokines, such as IFNγ (Figure 1). Like so many other inflammatory cytokines, IL-15 also induces counter-regulatory mechanisms to avoid overwhelming IL-15-mediated inflammation. For instance, IL-15 given continuously to human NK cells in vitro results in enhanced proliferation but decreased cytolysis, decreased production of IFNγ, decreased killing of liquid tumors in vivo, and increased cell death, all due (in part) to impaired fatty acid oxidation [57]. The membrane-bound A disintegrin and metalloprotease 17 (ADAM17), activated downstream of IL-15 signaling, restrains proliferation of NK cells in another example of negative feedback [58]. These data may help explain the findings of Sones, in which inappropriately high levels of IL-15 in preeclamptic mice are associated with reduced numbers of uNK cells [48].

Recent studies have also shed new light on how uNK interpret signals from IL-15 to carry out numerous functions essential to pregnancy, such as the release of angiogenic factors and growth factors, driving spiral artery remodeling, and shaping trophoblast invasion [59,60,61]. Sliz showed that the scaffolding protein GRB2-associated binding protein 3 (Gab3) was required for expansion of NK cells in response to both IL-15 (and IL-2) through CD122/γc [62]. Gab3-deficient NK cells stimulated with IL-15 exhibited major defects in phosphorylation of the mitogen-activated protein (MAP) kinases extracellular signal-related kinase (ERK), p38, and c-Jun N-terminal kinase (JNK) but phosphorylated signal transducer and activator of transcription 5 (STAT5) and Akt normally (Figure 1). Consistent with impaired IL-15 responsivity, NK cells in the uterus are reduced in the absence of Gab3, although only conventional NK1.1+ NK cells, not DBA+ NK cells, were affected. Regardless of the subset affected, abnormal spiral artery remodeling and trophoblast invasion were observed in Gab3-deficient dams, again supporting the importance of IL-15 in healthy pregnancy.

Recent human data show that a subset of decidual NK cells lacking expression of killer cell immunoglobulin-like receptors (KIRs) and CD39 proliferate strongly in response to IL-15 [63]. KIR-CD39- NK cells stimulated with IL-15 then acquire expression of KIRs and CD39, a phenotype previously identified as NK1 and enriched for transcripts encoding the cytolytic mediators perforin and granzymes by scRNAseq [13]. These data indicate that IL-15 supports the differentiation and function of NK cells in the uterus. These data also provide evidence that responsiveness to IL-15 among subsets in the uterus is heterogeneous. Marinić demonstrated that culture of primary NK cells in IL-15 or in conditioned medium from ESFs promotes migration of NK cells into Matrigel [25]. Knockdown of *IL15* in ESFs with siRNA or treatment of conditioned medium with anti-IL-15 antibody led to a marginal decrease in NK cell migration, raising the possibility that not all migration by NK cells in response to conditioned ESF medium is due to IL-15. Future work directed at how uNK subsets differentially integrate signals from IL-15 will be useful to better understand IL-15-mediated adverse outcomes of pregnancy, as discussed above.

While killer lymphocytes are the canonical responders to IL-15 by virtue of high expression of CD122 and γc, several other cells expressing IL-15 receptor components have been identified at the maternal–fetal interface. We recently showed that a subset of macrophages (Macs) in the uterus of mice and humans expresses CD122 [64]. CD122+Macs are enriched for interferon-stimulated transcripts and respond to type I and II IFNs by inducing CD122 in a dose-dependent manner. These Macs respond biochemically to stimulation with the IL-15/IL-15Rα complex by phosphorylating ERK and respond functionally by enhancing the release of proinflammatory cytokines (Figure 1). Unique tools to understand roles for CD122 on Macs have been developed and are being studied in our laboratory to better understand how IL-15-responsive macrophages impact pregnancy.

In addition to Macs, trophoblasts and trophoblastic cell lines have been found to express CD122 and respond to IL-15. Derived from choriocarcinomas, the cell lines JEG-3, BeWo, and JAR all expressed the *IL2RB* transcript by RT-PCR, but protein expression was not examined [65]. IL-15 increases the invasion of JEG-3 cells into Matrigel in a dose-dependent fashion at 1 and 10 ng/mL of recombinant IL-15 [66]. At 10 ng/mL of IL-15, JEG3 invasion increased by a factor of two, relative to untreated JEG3 cells. IL-15 does not affect proliferation of JEG-3 cells at any dose, as measured by activity of mitochondrial dehydrogenases. Similar data were obtained recently using HTR8 cells, which are immortalized first-trimester trophoblasts [25]. Finally, a modest increase in secretion of matrix metalloproteinase 1 (MMP1), but not MMP2 or MMP9, by JEG3 cells was shown by ELISA in response to culture in the presence of IL-15 [66]. Interpretation of these data derived from JEG3 cells is complicated by how distantly related JEG3 cells and primary EVTs are at a global transcriptional level, with almost 1200 genes significantly different by microarray between the two cell types [67].

In primary cells obtained from 8 to 12 weeks gestation, Apps compared the transcriptomes of flow-sorted EGFR+ VTs and HLA-G+ EVTs cultured for 12 h on fibronectin. While this method of obtaining EVT may be criticized, cross-referencing this dataset with that of the recently published scRNAseq data from the first-trimester maternal–fetal interface [13] confirmed that the vast majority of genes associated with EVT or VT in the Apps dataset were indeed associated specifically with EVT or VT (data not shown). *IL2RB* was highly enriched in EVTs [67]. Strong extracellular staining of CD122 protein was evident by flow cytometry on EVTs (Figure 1), with weak but non-zero expression by VTs. Expression of CD122 was also seen by IHC in a column of EVTs outside of a villous, with minimal staining in the villous itself.

Interestingly, γc is not detected in EVT or VT (Figure 1) [13,67]. It has been established in cell lines that CD122 can homodimerize [68,69]. Ferrag showed in CHO cells that chimeric receptors, containing the intracellular domain of CD122 paired with unique extracellular domains, signaled through JAK2 and STAT5 [68]. Co-transfection of COS-7 cells with differentially tagged human CD122, followed by co-immunoprecipitation, showed that CD122 can homodimerize [69]. CD122 prefers to heterodimerize with γc, if γc is available. With fluorescence resonance energy transfer (FRET), Pillet showed that the N-terminal (extracellular) domain of CD122 was absolutely required for homodimerization, while the intracellular domains were dispensable for homodimerization. With flow cytometry using tagged IL-2, CD122 homodimers bound IL-2 that could be blocked by an anti-CD122 antibody. Finally, cells expressing CD122 homodimers bound IL-2 with a similar, intermediate affinity as cells expressing heterodimers of CD122 and γc. Altogether, these data suggest that CD122 homodimers bind IL-2, but it is unclear how these homodimers interact with IL-15 in the presence or absence of IL-15Rα. Further, it remains unknown how IL-15 affects the function of trophoblasts, especially EVTs that invade into an IL-15-rich decidua.

Expression of CD122 on trophoblasts is unusual in several ways. Cohen found a long terminal repeat (LTR) with endogenous retroviral promoter elements 25kb upstream of the native promoter for *IL2RB* [70]. This LTR contains a transcriptional start site (TSS) and can splice into the normal ATG residing in exon 2 of *IL2RB*. RT-PCR for the LTR-containing transcript containing the retroelement showed that this alternative transcript was abundant in placenta (nearly 90% of transcripts) and rare in spleen and other organs, including peripheral blood mononuclear cells (PBMCs). Using a luciferase assay, a construct containing a portion of the LTR upstream of the splice donor site drove luciferase expression strongly in the choriocarcinoma cell line JEG-3. In vitro methylation of the luciferase construct eliminated luciferase expression under control of the LTR. Supporting the notion that epigenetic factors unique to trophoblasts allowed the LTR to drive expression of *IL2RB*, 9 C followed by G dinucleotide (CpG) sites within the LTR were identified and found to be consistently methylated in PBMCs but methylated less frequently in placenta and enriched trophoblasts. Inhibition of histone deacetylases to improve accessibility of the LTR also enhanced expression of *IL2RB* in cell lines. At the protein level, extracts of bulk placental villi were probed with two C-terminal-specific antibodies by Western blot. Full-length 75 kDa CD122 was not detected with these antibodies, only a 37 kDa fragment that may correspond to a cleaved fragment of CD122, something that has been previously demonstrated in leukemic cell lines [71]. Sequencing was not carried out to confirm the identity of this fragment, nor was subcellular localization of CD122 to confirm that it was membrane bound and not localized elsewhere. Further, the lack of full-length CD122 is at odds with flow cytometry data demonstrating clear expression of CD122 on trophoblasts [67]. If CD122 is translated and cleaved subsequently in trophoblasts, the N- and C-terminal fragments may have distinct and novel functions that have yet to be explored.

## 7. Conclusions

IL-15 is abundant and tightly regulated in the uterus during pregnancy. Several lines of evidence suggest that homeostasis of IL-15 must be maintained to ensure the health of the mother and fetus. Given widespread interest in IL-15 as a mediator of anti-tumor immunity in preclinical studies, numerous biological agents have been synthesized to mimic or inhibit the activity of IL-15 on target cells [51,72]. IL-15 agonists succeed in expanding killer lymphocytes but fail to reject tumors in humans when administered alone. Thought to be due to compensatory induction of anti-inflammatory signaling by IL-15, these observations may be of great interest in the context of reproduction, which depends on appropriate inflammation that maintains tolerance to the fetus. With a deeper understanding of the spatial and temporal regulation of IL-15, as well as its effects on classical and novel cell types, we can better detect dysregulation of IL-15 and associated immunopathology during gestation. We may then modulate IL-15 signaling in the uterus to optimize outcomes of pregnancy.

## Figures and Tables

**Figure 1 ijms-22-11094-f001:**
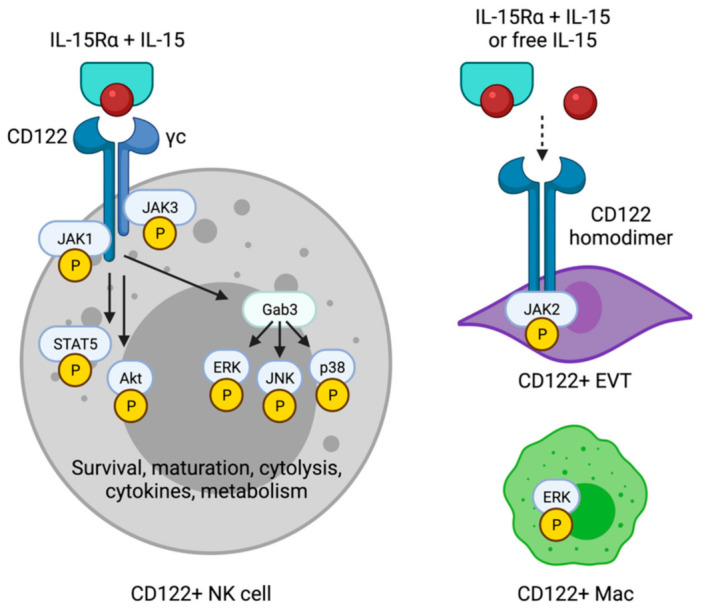
Components of IL-15 receptors and signaling cascades in IL-15-responsive cell types at the maternal–fetal interface. NK cells express CD122 and the common gamma chain (γc) and activate JAK-STAT, Akt, and MAP kinases in response to IL-15 presented in the context of IL-15Rα. IL-15 drives numerous functions in killer lymphocytes, including classical and uterine NK cells. Emerging IL-15-responsive cell types, such as CD122+ macrophages (CD122+ Mac) and CD122+ extravillous trophoblasts (EVT), may express alternative forms of the IL-15 receptor and activate non-classical signaling cascades downstream of IL-15. Created with Biorender.com, accessed 3 October 2021.

**Figure 2 ijms-22-11094-f002:**
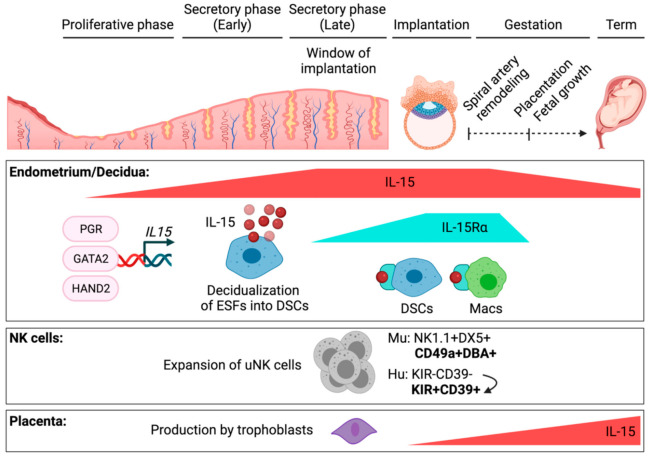
Spatial and temporal regulation of IL-15 during the menstrual cycle and during gestation has multiple effects during pregnancy. IL-15 is produced and presented by decidual stromal cells (DSCs) and macrophages (Macs) in the uterus. *IL15* is induced during the late secretory phase, when progesterone is dominant. Transcription of *IL15* is induced by the progesterone receptor (PGR) and GATA2. HAND2 binds the *IL15* promoter directly and induces *IL15* in DSCs but represses it in endometrial stromal fibroblasts, precursors to DSCs. Drawing on murine data, IL-15Rα peaks post-implantation, at the time of uterine spiral artery remodeling. The presumed abundance of IL-15/IL-15Rα complexes during that time expands murine DBA+ uterine NK cells and drives differentiation of KIR+CD39+ NK cells in human decidua. In the placenta, trophoblasts can produce IL-15 early in gestation at low levels but produce IL-15 maximally late in gestation during spontaneous labor. Created with Biorender.com, accessed 3 October 2021.

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
