# Peer review of "Interleukin-15 in Outcomes of Pregnancy"

_ijms, 2021, doi:10.3390/ijms222011094_

Round 1

Reviewer 1 Report

Gordon nicely describes a role of IL-15 in the uterus at pregnancy in this review. Since the review is comprehensive and up-to-date, it should provide readers with recent understanding of how fetuses are tolerated in the uterus during pregnancy. I would like to point two minor issues.

  1. There are some sentences that are not logically connected with other sentences. For example, a sentence “Levels of Il15ra and IL-15Ralpha protein were not examined” in lines 277-278, page 7 is suddenly appeared. Please read the text carefully and edit it.

  1. Since numbers of abbreviations are used in this review, it is hard for a reader who is not familiar with the field like me to follow the story. I suggest the author to add an abbreviation section.

Author Response

Thank you for the favorable review and for your constructive comments.  All changes are highlighted in yellow.

1. There are some sentences that are not logically connected with other sentences. For example, a sentence “Levels of Il15ra and IL-15Ralpha protein were not examined” in lines 277-278, page 7 is suddenly appeared. Please read the text carefully and edit it.

In response to this comment, language was edited, and several sentences were shortened or removed to make this section less clumsy.  Please see the new paragraph from lines 272-285. The specific sentence mentioned was unnecessary and was deleted.

2. Since numbers of abbreviations are used in this review, it is hard for a reader who is not familiar with the field like me to follow the story. I suggest the author to add an abbreviation section.

An abbreviation section was added as requested.  I placed it after the conflict of interest statement, lines 459-473.  Additional abbreviations were defined explicitly in the text (highlighted) whenever possible.

Reviewer 2 Report

The authors summerized novel data on IL-15 function in reproduction and elucidated IL-15 cooperation with other factors mediating normaldevelopment of pregnancy. 

It is a very well done review. It does not requirecorrection and further revision.

Author Response

Thank you for the favorable review. Specific changes suggested by Reviewer 1 are highlighted in yellow.